# Longitudinal impact of screening colonoscopy on greenhouse gas emissions

**Hasiya Yusuf[1], Vinita Gupta[2], Ikponmwosa Osaghae[3], Abhishek Kumar[2,4]***

**1** Department of Internal Medicine, Jacobi Medical Center, Albert Einstein College of Medicine, Bronx, New York, United States of America, **2** Division of Hematology &Oncology, Department of Internal Medicine, Jacobi Medical Center, New York City Health and Hospital Corporation, Bronx, New York, United States of America, **3** Department of Epidemiology, MD Anderson Cancer Center, The University of Texas, Houston, Texas, United States of America, **4** Division of Hematology & Oncology, Department of Internal Medicine, Albert Einstein College of Medicine, Bronx, NY, United States of America

* research.ak@gmail.com

**Data Availability Statement:** No specific data was used in this study. All information used in the draft were derived from published articles. The citations for all articles are provided in the manuscript.

## Abstract

### Background and aim

Colonoscopy is the gold-standard screening test for colorectal cancer. However, it has come under scrutiny for its carbon footprint and contribution to greenhouse gas (GHG) emissions compared to other medical procedures. Notwithstanding, screening colonoscopies may have a positive effect on GHG emissions that is unknown. This study estimated the carbon emissions prevented by screening colonoscopies in the U.S.

### Methods

Using the reported number of screening colonoscopies performed annually in the U.S. and the absolute risk reduction (ARR) reported in the NorDICC trial, we calculated the expected minimum number of cancer treatment and surveillance visits prevented through screening based on the cancer stage. The average carbon emission averted per mile traveled was computed using the Environmental Protection Agency's (EPA) GHG equivalencies calculator. The final estimate of carbon emissions averted over a decade by screening colonoscopies performed in one year was determined.

### Result

6.3 million screening colonoscopies performed in one year prevent 1,134,000 colorectal cancers over a ten-year period. Of these, 38·3% (434,254) are localized, 38·8% (440,281) are regional, and 22·9% (259,465) are metastatic disease. The minimum number of post-diagnosis visits prevented is 11 for stage I, $\geq 21$ for stage II, $\geq 25$ for stage III, and $\geq 20$ for stage IV disease, comprised of diagnostic, surgical evaluation, chemotherapy, and surveillance visits. The total number of visits prevented by screening is 2,388,397 for stage I, 5,254,421 for stage II, 13,120,369 for stage III, and 9,210,972 for stage IV disease. Approximately 395 million miles of travel and 158,263 metric tons of CO2, equivalent to 177 million pounds of coal burned, 19 billion smartphones charged, or 18 million gallons of gasoline consumed, were saved over ten years through screening.

**Funding:** The author(s) received no specific funding for this work.

**Competing interests:** I have read the journal's policy and the authors of this manuscript have the following competing interests: Abhishek Kumar has stocks and stock options in Abbvie, AVEO, Cara Therapeutics, Celgene, Iovance Biotherapeutics, Eli Lilly, Viking Therapeutics, SPDR S and P Biotech ETF, Agenus, Amgen, BioTelemetry, Bristol-Myers Squibb, Bio-Path Holdings, Inc, ChemBio Diagnostic Systems, CRISPR therapeutics, CVS Health, Editas Medicine, Five Prime Therapeutics, Immunomedics, Livongo, Medtronic, Northwest Biotherapeutics, PTC Therapeutics, Regeneron, Teladoc, Trovagene, Vertex, Globus Medical, Acadia Pharmaceuticals, ADMA Biologics, BeyondSpring Pharmaceuticals, Cardiff Oncology, IDEXX Laboratories, Alkido Pharma, Albireo Pharma, Blueprint Medicines, Precision, Biosciences, Novavax, Poseida Therapeutics, Surgalign, AstraZeneca, Contrafect, CryoLife, Geron, Johnson & Johnson/Janssen, Kronos, Ontrak, Spectrum Pharmaceuticals, Uniqure, Renalytix AI PLC, Sierra Oncology, Vericel, Viatris, CUE Biopharma, DermTech, Gevo, Jazz Pharmaceuticals, Purple Biotech, Protagonist Therapeutics, Schrodinger, and Sensei Biotherapeutics. He also received travel and hotel funding for a lecture from the American Society of Clinical Oncology. All other authors have no competing interests.

## Conclusion

Colorectal cancer screening decreases cancer-related GHG emissions and minimizes the environmental impact of cancer treatment.

## Background

Colorectal cancer is the third leading cause of cancer death in the United States (U.S.) and the second most common cause of cancer-related death when men and women are combined [1]. Based on projections by the American Cancer Society (ACS), 153,020 persons will be diagnosed with colorectal cancer in 2023, and 52,550 will die from the diagnosis, many of whom will be under 50 years of age [2]. Colonoscopy is considered the gold standard screening test for colorectal cancer and has become both popular and increasingly acceptable, especially in developed countries. In the U.S. alone, approximately 15 million colonoscopies are performed annually, of which 6·3 million are screening colonoscopies [3, 4].

Notwithstanding, colonoscopy has been scrutinized for its carbon footprint and contribution to greenhouse gas (GHG) emissions. This concern arises from mounting evidence of the enormous contribution of the healthcare sector to GHG emissions and as part of global efforts toward curtailing the rapid impact of human activities on the climate. It has been shown that the healthcare sector is responsible for 5% of global GHG emissions [5]. To put this in perspective, if the healthcare sector were a country, it would be the fifth largest emitter of GHG. Indirect emissions are the most significant source of healthcare-associated GHG emissions, accounting for 50–75% of all healthcare emissions [5]. Disposable materials, such as those utilized in endoscopic procedures, fall under this category. Gastrointestinal endoscopic procedures are the third highest source of medical waste in healthcare facilities and a significant contributor to the carbon footprint [3, 6].

In a study of endoscopic procedures performed in two U.S. medical institutions by Namburar et al., [7] a single endoscopic procedure resulted in 2·1 kg of disposable waste, of which only nine percent was recycled. Similarly, Gayam found that each endoscopic procedure generated an estimated 1·5 kilograms of plastic waste [8]. Notwithstanding, the necessity of screening colonoscopies for cancer prevention remains undeniable. In addition, screening colonoscopies may have a longitudinal positive effect on GHG emissions and the climate we may have overlooked. By reducing colorectal cancer incidence and averting the need for cancer treatment, colorectal cancer screening may reduce the carbon footprint of healthcare. However, no study has estimated the GHG emissions, carbon footprint prevented, or colonoscopy's net impact on the climate. In this study, we calculated the carbon emissions prevented by screening colonoscopies performed in one year in the U.S.

## Methods

We determined the number of screening colonoscopies performed annually in the U.S. from published studies and national statistical databases [3, 4]. The absolute risk reduction (ARR) of colorectal cancer incidence in patients who received screening colonoscopies was derived from findings of the NordICC trial [9]. The NordICC trial was conducted across four European countries (Poland, Sweden, Norway, and the Netherlands). Persons aged 55–64 years with no prior colonoscopy screening were enrolled and randomly assigned into two groups: Group One, who received screening colonoscopy, and Group Two, who were not invited for

screening colonoscopy (routine screening was not standard of care in the study regions at the time of the study) in a 1:2 ratio. A total of 68,208 participants were enrolled, of which 11,843 were screened and 56,365 did not receive screening. The incidence of colorectal cancer at a median follow-up of ten years was estimated. Using the ARR reported in the study and the Center for Disease Control's (CDC) data on new cancer diagnoses, we estimated the total number of colorectal cancer diagnoses prevented through screening by stage at diagnosis (localized (I and II), regional (III), and distant or metastatic disease (IV)) [10]. No studies reporting the proportion of patients diagnosed with stage I versus stage II disease were identified. Hence, to derive the number of colorectal cancers averted for these two cancer stages, we assumed that 50% of patients with localized disease had either stage I or II.

Using the National Comprehensive Cancer Network's (NCCN) guidelines for the treatment and surveillance of colon cancer, we determined the average number of surgical consultations and diagnostic, treatment, radiology, and surveillance visits by cancer stage. Based on the guidelines, all patients should, at the time of diagnosis, undergo an initial colonoscopy with biopsy, surgical evaluation, laboratory workup, and imaging with chest CT and abdominal CT to evaluate for metastasis. We assumed a minimum of three visits for surgical evaluation (one each at pre-operative, operative, and postoperative follow-up) in patients with stage I to III disease. Treatment recommendations for stage IV disease are highly variable and complex. They depend on the site and number of metastatic lesions, tumor resectability, conversion to resectable tumor after initial treatment, patient's functional status, presence or absence of mutations, etc. For simplification, we assumed all stage IV colorectal cancers were unresectable, and mutations were absent. As a result, surgical evaluation visits were excluded from visit estimates. Surveillance visits were also excluded, and treatment visits were limited to one year based on the median survival of patients with stage IV cancers.

Pre-diagnoses visits, laboratory investigations, and physician visits outside of scheduled surveillance and treatment visits were excluded from the analysis due to the challenges with calculations posed by variations in the frequency of visits required for each patient, often determined by patient complexities and specific needs. The minimum number of diagnostic, surgical evaluation, treatment, and surveillance visits for stages I-IV were determined. The carbon footprint of colorectal cancer screening was derived using the average distance traveled by patients with colorectal cancer to receive care and the average carbon emission from these travels. The expected average distance traveled by each patient to receive care utilized in this study is 13·2 miles, based on the estimates from the Healthcare Cost and Utilization Project's (HCUP) data derived from patients across different geographical regions of the US. [11]. Average carbon emission of 400 grams of CO2 per mile traveled was used based on the Environmental Protection Agency's (EPA) online greenhouse gas equivalencies calculator [12]. The final estimate of carbon emissions averted through screening colonoscopies are presented in metric tons, pounds of coals burned, number of smartphones charged, and gallons of gasoline consumed.

Estimates of colorectal cancers averted through screening are based on the reported 6·3 million (42% of the 15 million total annual colonoscopies) performed in the U.S. [4]. Calculation of visits are approximated to the nearest whole numbers. Approval was not required or obtained from the institutional review board for this study as all study data and information were from publicly available sources or previously published research articles.

## Result

Using a total annual screening colonoscopy of 6·3 million and an ARR of 0·18, [9], 1·134 million colorectal cancers are prevented over a decade by one year of colorectal cancer screening.

Of these, 35·9% (407,106) are localized, 36·4% (412,776) are regional, 21·4% (242,676) have distant metastasis, and 6·3% (71,442) are of unknown stage [10]. Assuming that cancers of unknown staging have a similar distribution as stages I-IV cancers, of the total 71,442 colorectal cancers of unknown staging, 38·0% (27,148) would be localized or stages I and II, 38·5% (27,505) would be regional or stage III, and 23·5% (16,789) would be metastatic or stage IV disease. In total, 434,254 localized or stage I and II, 440,281 regional or stage III, and 259,465 metastatic or stage IV cancers are averted over a ten-year period by one year of colorectal cancer screening.

## Number of visits by cancer stage

For stage I disease, patients will undergo one diagnostic visit, a minimum of three visits for surgical evaluation (one pre-operative, operative, and postoperative follow-up), and seven surveillance visits (one annual visit for a total of five years, colonoscopy visit one-year post surgery, and a repeat colonoscopy in three years if advance adenoma is absent during the previous colonoscopy screening). Current treatment guidelines do not recommend treatment for stage I disease. Hence, patients with stage I colorectal cancer make 11 total visits.

Patients with stage II disease also undergo one diagnostic visit and a minimum of three surgical evaluation visits (one each at pre-operative, operative, and postoperative follow-up). Treatment of stage II colon cancers varies by guidelines and is guided majorly by the presence or absence of high-risk features on histology [13]. An estimated 80% of stage II cancers are low-risk and require no treatment. Only about 20% of stage II cancers are treated [14]. Two different standard regimens are used in the treatment of stage II colorectal cancers. Treatment with either intravenous fluorouracil, with or without oxaliplatin, or oral capecitabine, with or without oxaliplatin, for a duration of six months is recommended. Assuming that 50% of treatment-eligible patients received the first regimen and the other 50% received the second regimen, the former will have 24 treatment visits, and the latter will have eight visits (see **Table 1**). All diagnoses of stage II cancer undergo five years of surveillance, comprising ten follow-up visits, five imaging visits, and two colonoscopy visits (at least 17 surveillance visits). Thus, 80% of stage II cancers have 21 visits (one initial diagnostic visit, three surgical evaluation visits, and 17 surveillance visits). Of the 20% that are treatment eligible, the 50% that receive intravenous fluorouracil have 45 total visits (one diagnostic visit, three surgical evaluation visits, 17 surveillance visits, and 24 treatment visits). The other 50% who receive oral capecitabine make 29 visits in total (one diagnostic visit, three surgical evaluation visits, 17 surveillance visits, and eight treatment visits).

Similar to stages I and II, patients with stage III cancer have one diagnostic and a minimum of three surgical evaluation visits. Treatment is variable and guided by the presence or absence of high-risk features. Approximately 60% of all stage III cancers are low risk, while 40% are considered high risk [15, 16]. Low-risk stage III cancers are preferably treated with three months of oral capecitabine and intravenous oxaliplatin (four treatment visits) based on the outcome of the IDEA trial [16]. Patients with high-risk features (40% of patients diagnosed with stage III) are treated with either a six-month course of Capecitabine (eight treatment visits) or 5-fluorouracil (24 treatment visits). We assumed that 50% of patients received either 5-fluorouracil or Capecitabine. Like stage II, patients diagnosed with stage III get five years of surveillance, comprising ten follow-up visits, five imaging visits, and two colonoscopy visits (17 surveillance visits). Therefore, of the patients with stage III, 60% (low-risk group) have 25 visits (one diagnostic visit, three surgical evaluation visits, 17 surveillance visits, and four treatment visits), 20% (the high-risk group treated with Capecitabine) have 29 visits (one diagnostic visit, three surgical evaluation visits, 17 surveillance visits, and eight treatment visits) and 20%

**Table 1. Number of visits, patient travel, and carbon emissions prevented by colorectal cancer screening by cancer stage.**

| Cancer Stage | *Cancers Prevented by Screening | Diagnosis Visit | Surgical Evaluation Visit | Treatment Visit | Surveillance Visit | Total Visits by Treatment Type | Total Visits by Cancer Stage | Distance Travelled (Miles) | Carbon Emissions (Metric tons) |
|---|---|---|---|---|---|---|---|---|---|
| **Stage I** | 217,127 | 1 | 3 | 0 | 7 | 2,388,397 | 2,388,397 | 31,526,840 | 12,611 |
| **Stage II** | 217,127 | 1 | 3 | Low-risk untreated group (80%)– 0 | 17 | 3,647,721 | 5,254,421 | 69,358,357 | 27,743 |
| | | | | High-risk group treated with capecitabine based regimen (10%) - 8 | | 629,648 | | | |
| | | | | High-risk group treated with 5-fluorouracil-based regimen (10%) - 24 | | 977,040 | | | |
| **Stage III** | 440,281 | 1 | 3 | Low-risk group treated with three months of capecitabine-based regimen (60%) - 4 | 17 | 6,604,225 | 13,120,369 | 173,188,871 | 69,275 |
| | | | | High-risk group treated with 6-months of capecitabine-based regimen (20%) - 8 | | 2,553,624 | | | |
| | | | | High-risk group treated with six months of 5-fluorouracil-based regimen (20%) - 24 | | 3,962,520 | | | |
| **Stage IV** | 259,465 | 1 | 0 | Two-weekly regimen of 5-fluorouracil) (50%) - 52 | 0 | 6,875,796 | 9,210,972 | 121,584,830 | 48,634 |
| | | | | Three-weekly Capecitabine regimen (50%) - 17 | | 2,335,176 | | | |
| **Total** | 1,134,000 | 4 | 9 | 137 | 41 | | 29,974,159 | 395,658,898 | 158,263 |

*Estimated number of colorectal cancers prevented over a ten-year period by colorectal cancer screening in one year

(the high-risk group treated with 5-fluorouracil) have 45 visits (one diagnostic visit, three surgical evaluation visits, 17 surveillance visits, and 24 treatment visits).

All patients with stage IV diagnosis have one diagnostic visit and three to six-monthly imaging in the first year of treatment (three visits). Assuming a treatment duration of one year (median survival for stage IV colorectal cancer) [17], and that 50% of patients with stage IV cancer receive either a two-weekly regimen of 5-fluorouracil (two visits/cycle for 26 cycles or 52 visits) or three-weekly Capecitabine (17 visits), 50% have 18 visits (one diagnostic and 17 treatment visits), and 50% have 53 visits (one diagnostic visit and 52 treatment visits). **Table 1.** summarizes the number and types of visits by cancer stage.

## Total expected visits

Stages I and II constitute localized disease. Assuming that 50% of localized disease is stage I and 50% is stage II, half of the total number of visits for localized colorectal cancer (217,127) would be averted for stage I and II, respectively. Therefore, 217,127 x 11 visits or 2,388,397 visits are averted for stage I. As previously outlined above, 80% of patients with stage II cancers (that is, 80% of 217,127) have 21 visits, which results in a total of 3,647,733 visits. Of the 20% that are treatment eligible, 50% (21,712) treated with 5-fluorouracil based regimen have 45 x 21,712 (977,040) visits, and the other 50% who receive a capecitabine based regimen have 29 x 21,712 (629,648). The total number of visits averted overall for stage II disease is 5,254,421. Of

the 440, 281 stage III colorectal cancers prevented through screening colonoscopy, 60% (264,169) representing the low-risk group would have 264,169 x 25 (6,604,225) visits, 20% (88,056), or the high-risk Capecitabine treated group would have 88,056 x 29 (2,553,624) visits and 20% (88,056), or the high-risk 5-fluorouracil treated group would have 88,056 x 45 (3,962,520) visits. A total of 13,120,369 visits are averted for stage III disease. For stage IV, 259,465 colonoscopies were prevented. Of these, 50% (129,732) would have had 18 visits, and 50% (129,732) would have had 53 visits. Thus, 129,732 x 18 (2,335,176) and 129,732 x 53 (6,875,796) visits, respectively, or 9,210,972 visits, were prevented for stage IV. Based on these estimates, a total of 29,974,159 hospital visits are prevented by a single ten-yearly colorectal cancer screening. The calculation of expected visits is summarized **in Table 1.**

### Estimated carbon footprint

Using an estimated average median distance traveled to receive care of 13·2 miles across different geographical regions of the U.S. based on findings of the Healthcare Cost and Utilization Project's (HCUP) data, patients with stage I disease would have traveled 31,526,840 miles, 69,358,357 if stage II, 173,188,871 miles if diagnosed with stage III, and 121,584,830 miles if stage IV, averting 395,658,898 miles of travel. **Fig 1** compares carbon emissions from screening colonoscopies to both carbon emissions prevented through screening by cancer stage and total carbon emissions prevented. Per EPA's estimates, the average carbon emission of typical passenger vehicles is 400 grams of CO2 per mile. This amounts to $1.6 \times 10^{11}$ grams of CO2 or 158,263 metric tons of CO2 emissions prevented over ten years, equivalent to 177 million

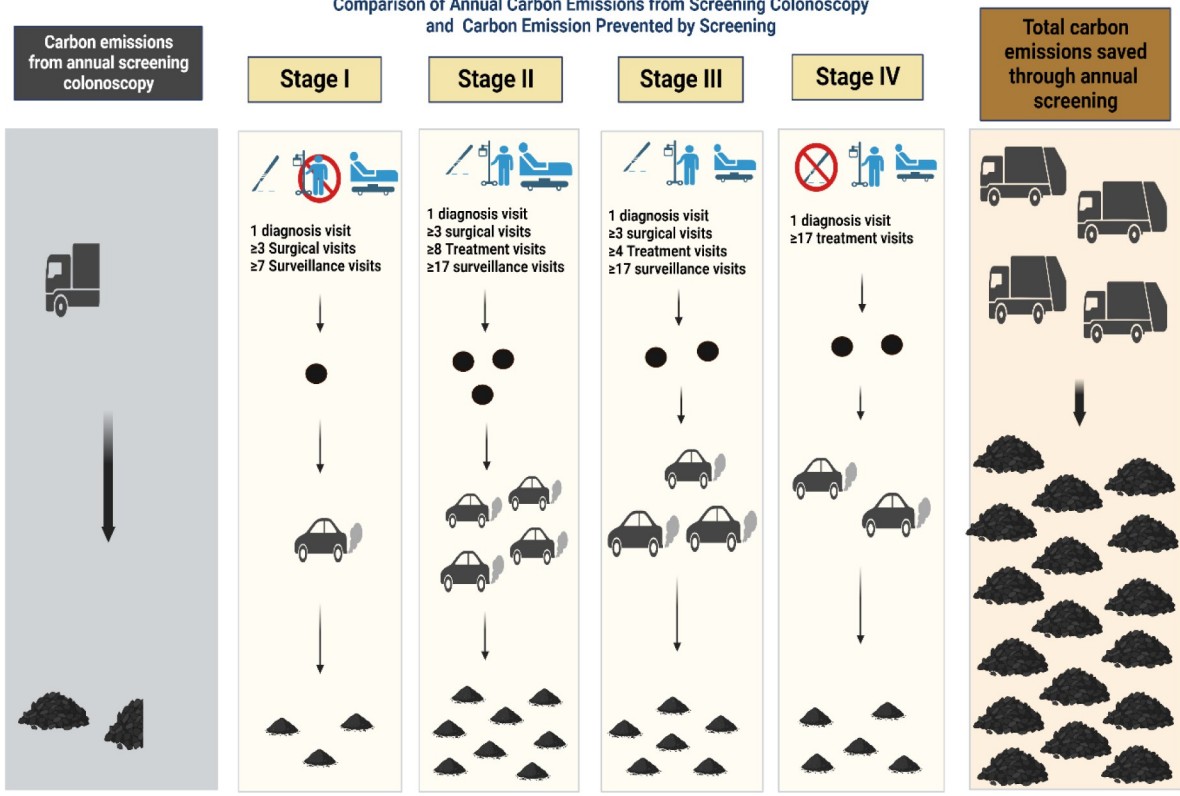

**Fig 1. Comparison of annual carbon Emissions from colorectal cancer screening, carbon emissions prevented by screening colonoscopy per patient by cancer stage, and carbon emission prevented by annual screening.**

pounds of coal burned, 19 billion smartphones charged, or 18 million gallons of gasoline consumed.

## Discussion

In this first-ever study to assess the carbon footprint averted through screening colonoscopies, we found that 395 million miles of travel and approximately 158,000 metric tons of CO2 emissions, equivalent to 177 million pounds of coal burned and 18 million gallons of gasoline consumed, are prevented over ten years by the 6·3 million annual screening colonoscopies performed in the U.S. This is nearly five times the 3·9 million gallons of gasoline or almost 40 million pounds of coal burnt generated from total colonoscopies performed in one year, and more than ten times the carbon emission from annual screening colonoscopies [8]. This is the first study highlighting the potential environment-saving effect of screening colonoscopies. Many studies and opinion papers [3, 8, 18] have highlighted the carbon footprint of screening colonoscopy on the environment. However, no publication has estimated the positive impact of the procedure on GHG emissions. The result of this study is crucial when considered in light of the pursuit of environmentally sustainable healthcare, which seeks to mitigate healthcare pollution and minimize healthcare-associated GHG emissions. Human activities contribute up to 40 billion metric tons of carbon dioxide to the atmosphere annually, and screening colonoscopies is considered a significant contributor to healthcare-associated GHG emissions [19]. However, our findings provide the first evidence that screening colonoscopy is potentially an environment-saving procedure that prevents future GHG by preventing future vehicular transport, which, at present, is an inseparable part of cancer care.

Understanding the GHG emission averted through screening colonoscopies provides a new perspective in the burgeoning conversations on the carbon footprint of healthcare practices, of which colonoscopy is a significant component. However, it must be emphasized that our findings do not underestimate or minimize the impact of colonoscopy on the environment or the need to adopt more conservative and environment-friendly approaches in colorectal cancer screening. Instead, it provides a holistic view of the effects of colonoscopies on the environment. Given our current understanding that colorectal cancer screening prevents future cancer-associated GHG emissions in addition to decreasing morbidity and mortality, we anticipate that ongoing efforts to reduce waste generation through the process of recycling and moving in the direction of what has been termed "green endoscopy" [20], will have an even more significant net advantage on the environment in the fight toward minimizing healthcare's influence on the climate.

Calculations of GHG emissions presented here are limited to carbon emissions from patient transport and do not consider other components of cancer treatment that equally affect the environment. As a result, our finding is likely an underestimation of the actual GHG emissions averted through screening. We also believe that the estimate of colorectal cancer incidence prevented through colonoscopies and the subsequent estimate of the total carbon footprint averted is modest compared to the actual picture **should all those eligible for colorectal cancer screening get screened.** In addition, although the carbon footprint of cancer has not been directly quantified, the supply of pharmaceuticals and medical devices is responsible for a third of healthcare-associated carbon emissions [5]. Therefore, by averting the incidence of colorectal cancer, screening has the added benefit of preventing GHG emissions from ancillary treatments.

This study is limited by the diverse complexities of cancer treatment in general and colorectal cancer treatment in particular. There are multiple chemotherapeutic and targeted therapeutic regimens used in varied combinations and permutations and discrepancies in the duration of treatments within and between cancer stages. Variability in treatment regimens for the same cancer also results from provider preferences and country or region of residence, making

precise estimates of treatment visits near impossible. Nonetheless, calculations were completed using the least possible number of visits, erring on the side of underestimation. Patient visits due to cancer-associated complications, such as ED visits or hospitalizations, were also excluded from the analysis. Only standard visits that are common to patients based on cancer stage were included in the analysis.

The absolute risk reduction of colorectal cancer utilized in calculating the number of colorectal cancers averted are estimates based on recent findings of a European research study. While the countries of study and the U.S. population to which we extrapolate these findings are both industrialized or developed nations, we acknowledge that there might be fundamental differences across these populations and that the incidence of colorectal cancer may differ. However, we are confident that these differences are minimal based on our extensive reviews that reflect an overall similarity in the incidence, prevalence, and mortality of colorectal cancer across these populations.

In addition, the primary study used to estimate colorectal cancer incidence and death averted enrolled persons aged 55–64 years, which excludes a considerable proportion of persons eligible for colorectal cancer screening, that is, ages 45–54 and ages 65–75 years. We acknowledge that these may have affected our overall final estimates of colorectal cancers averted. However, the estimate of cancers averted is likely higher than calculated, as the incidence of colorectal cancer has been on the rise in the 45–54 age group.

The estimate of annual colonoscopies used here was based on the numbers reported for the U.S. population. Estimates will vary for other countries based on the prevalence of colorectal cancer screening and population size. Hence, findings may not be generalizable to populations significantly different from the U.S.

Finally, carbon emission was calculated based on EPA's estimates for passenger vehicles and may differ based on mode of transportation (personal versus public transport). We, however, believe that these differences are minimal.

In conclusion, screening colonoscopies has a net positive effect on the environment as it prevents the environmental impact of future colorectal cancer diagnoses. This effect can be heightened by emphasizing and adopting new strategies that reduce waste generation and consequent GHG emissions that accompany the procedure.

## Author Contributions

**Conceptualization:** Abhishek Kumar.

**Data curation:** Hasiya Yusuf, Abhishek Kumar.

**Formal analysis:** Abhishek Kumar.

**Methodology:** Hasiya Yusuf, Ikponmwosa Osaghae, Abhishek Kumar.

**Project administration:** Abhishek Kumar.

**Supervision:** Abhishek Kumar.

**Writing – original draft:** Hasiya Yusuf, Vinita Gupta, Abhishek Kumar.

**Writing – review & editing:** Hasiya Yusuf, Vinita Gupta, Ikponmwosa Osaghae, Abhishek Kumar.

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
