## [Decision Letter · Decision Letter 0]

21 Jun 2024

PONE-D-24-11124Longitudinal Impact of Screening Colonoscopy on Greenhouse Gas EmissionPLOS ONE

Dear Dr. Kumar,

Thank you for submitting your manuscript to PLOS ONE. After careful consideration, we feel that it has merit but does not fully meet PLOS ONE’s publication criteria as it currently stands. Therefore, we invite you to submit a revised version of the manuscript that addresses the points raised during the review process.

We look forward to receiving your revised manuscript.

Kind regards,

Alejandro Piscoya

Academic Editor

PLOS ONE

Journal Requirements:

Reviewers' comments:

Reviewer's Responses to Questions

**Comments to the Author**

1. Is the manuscript technically sound, and do the data support the conclusions?

Reviewer #1: Yes

Reviewer #2: Yes

2. Has the statistical analysis been performed appropriately and rigorously? 

Reviewer #1: I Don't Know

Reviewer #2: Yes

3. Have the authors made all data underlying the findings in their manuscript fully available?

Reviewer #1: Yes

Reviewer #2: Yes

4. Is the manuscript presented in an intelligible fashion and written in standard English?

Reviewer #1: Yes

Reviewer #2: Yes

5. Review Comments to the Author

Reviewer #1: 1. I wonder why the authors decided to narrow the target population from 45-77 to 55-65. The authors briefly touched base with the idea that the actual amounts would be different since they restricted the age period of the subjects. But I still do not understand why a narrowed age scale.

2. To calculate the carbon footprint the authors calculated the overall burden of the diagnosis of cancer which is reasonable. However, I wonder if they included the follow up colonoscopies after the first initial screening in the analysis. To claim that only a single-initial colonoscopy prevents all those cancers is too simplistic. We need to include all the follow-up screenings as well to see the actual benefit of colonoscopy.

3. Hopefully, as a next step the authors would compare stool-based screenings with endoscopic procedures.

Reviewer #2: Dear authors,

Just to comment that on page 5 that group 2 of patients included in the NordICC trial was not offered screening... This seems unethical to me; those individuals were deprived of the opportunity for detection.

6. PLOS authors have the option to publish the peer review history of their article (what does this mean?). If published, this will include your full peer review and any attached files.

Reviewer #1: **Yes: **Akif Altinbas

Reviewer #2: No

---

## [Author Response · Author response to Decision Letter 0]

23 Jun 2024

Title of Paper: “Longitudinal Impact of Screening Colonoscopy on Greenhouse Gas Emission”

Initial decision: Revise & resubmit

Corresponding Author: Dr. Abhishek Kumar, MD

Dear Dr., Chenette,

On behalf of myself and my co-authors, I submit the enclosed revised manuscript titled “Longitudinal Impact of Screening Colonoscopy on Greenhouse Gas Emission” for publication in your journal. We appreciate the opportunity to revise our submission and to make relevant changes as recommended. We look forward to your review of our revised submission. Thank you for your invaluable feedback and for the opportunity.

General comments

1. I found your article to be very well-written, original, and highly relevant, given the current times. It is very well-structured and thought-out.

Response: We thank the reviewer for the kind feedback

2. On page 7, you mention that approval was not required... Why is that? 

Response: Approval was not required as data and other information presented in the manuscript were publicly available or previously published. This has been clarified in the manuscript to read, “Approval was not required or obtained from the institutional review board for this study as all study data and information were from publicly available sources or previously published research articles”

3. On page 14, where the study limitations are listed, we should also consider transfers due to treatment complications (hospitalizations, extra trips to be evaluated by their doctors for the presence of uncommon symptoms, etc.). 

Response: We thank the reviewers for pointing this out. In response, we have added the sentence, “Patient visits due to cancer-associated complications, such as ED visits or hospitalizations, were also excluded from the analysis. Only standard visits that are common to patients based on cancer stage were included in the analysis”.

Reviewer #1 comments

1. I wonder why the authors decided to narrow the target population from 45-77 to 55-65. The authors briefly touched base with the idea that the actual amounts would be different since they restricted the age period of the subjects. But I still do not understand why a narrowed age scale.

Response: We limited the target population to 55-65 years because this is the study population that was used in the Nordicc study. We have no study-based colonoscopy screening data for ages 45-77 years, although we acknowledge that this would have been a more appropriate age category. 

2. To calculate the carbon footprint the authors calculated the overall burden of the diagnosis of cancer which is reasonable. However, I wonder if they included the follow up colonoscopies after the first initial screening in the analysis. To claim that only a single-initial colonoscopy prevents all those cancers is too simplistic. We need to include all the follow-up screenings as well to see the actual benefit of colonoscopy.

Response: We thank the reviewer for this important observation. As noted, only participants who received a first initial screening were included in the analysis of the Nordicc trial. Similarly, only estimates of screening colonoscopies were used to calculate the overall burden of the diagnosis, with the exclusion of estimates from surveillance screenings. The decision to exclude surveillance colonoscopies was made to ensure uniformity between the methodology of the Nordicc trial and our current paper. We, however, acknowledge that including surveillance colonoscopies will provide more robust estimates that may be closer to the true estimates. However, as emphasized in our limitations and methodology, the goal was to err on the side of underestimation. In addition, our team is considering a further exploration of cancer diagnosis from screening. This study will assess both surveillance and routine colonoscopies alike. 

3. Hopefully, as a next step the authors would compare stool-based screenings with endoscopic procedures.

Response: We agree that it would be important to compare stool-based screenings with endoscopic procedures. We hope to examine this in a separate study in the future. 

Reviewer #2 Comments

1. Just to comment that on page 5 that group 2 of patients included in the NordICC trial was not offered screening... This seems unethical to me; those individuals were deprived of the opportunity for detection.

Response: Please note that patients in group two were not offered screening as colorectal cancer screening was not the standard of care in trial regions at the time. We have included the sentence “……. (routine screening was not standard of care in the study regions at the time of the study) for clarification. 

As a reminder, all authors have seen and approved the submitted and revised manuscript, please feel free to contact me through e-mail at research.ak@gmail.com. Thank you again for taking the time to review our revised manuscript.

Sincerely, 

Abhishek Kumar, M.D

Assistant Professor of Hematology and Oncology, 

Albert Einstein College of Medicine/Jacobi Medical Center

Building 1, 3N20

1400 Pelham Parkway S,

Bronx, N.Y – USA

Email: research.ak@gmail.com

---

## [Editor Report · Decision Letter 1]

2 Jul 2024

Longitudinal Impact of Screening Colonoscopy on Greenhouse Gas Emission

PONE-D-24-11124R1

Dear Dr. Kumar,

We’re pleased to inform you that your manuscript has been judged scientifically suitable for publication and will be formally accepted for publication once it meets all outstanding technical requirements.

Kind regards,

Alejandro Piscoya

Academic Editor

PLOS ONE
---

## [Editor Report · Acceptance letter]

11 Jul 2024

PONE-D-24-11124R1 

PLOS ONE

Dear Dr. Kumar, 

I'm pleased to inform you that your manuscript has been deemed suitable for publication in PLOS ONE. Congratulations! Your manuscript is now being handed over to our production team.

Kind regards, 

on behalf of

Professor Alejandro Piscoya 

Academic Editor

PLOS ONE